# Carcinoembryonic antigen-expressing oncolytic measles virus derivative in recurrent glioblastoma: a phase 1 trial

Evanthia Galanis [1,2] ✉, Katharine E. Dooley [3], S. Keith Anderson [3],
Cheyne B. Kurokawa[2], Xiomara W. Carrero[3], Joon H. Uhm[4], Mark J. Federspiel[2],
Alexey A. Leontovich[3], Ileana Aderca[2,8], Kimberly B. Viker[2], Julie E. Hammack[4],
Randolph S. Marks[1], Steven I. Robinson[1], Derek R. Johnson[5],
Timothy J. Kaufmann[5], Jan C. Buckner[1], Daniel H. Lachance[4], Terry C. Burns [6],
Caterina Giannini [7], Aditya Raghunathan[7], Ianko D. Iankov[2] & Ian F. Parney[6]

Measles virus (MV) vaccine strains have shown significant preclinical anti-tumor activity against glioblastoma (GBM), the most lethal glioma histology. In this first in human trial (NCT00390299), a carcinoembryonic antigen-expressing oncolytic measles virus derivative (MV-CEA), was administered in recurrent GBM patients either at the resection cavity (Group A), or, intratumorally on day 1, followed by a second dose administered in the resection cavity after tumor resection on day 5 (Group B). A total of 22 patients received study treatment, 9 in Group A and 13 in Group B. Primary endpoint was safety and toxicity: treatment was well tolerated with no dose-limiting toxicity being observed up to the maximum feasible dose ($2 \times 10^7$ TCID50). Median OS, a secondary endpoint, was 11.6 mo and one year survival was 45.5% comparing favorably with contemporary controls. Other secondary endpoints included assessment of viremia, MV replication and shedding, humoral and cellular immune response to the injected virus. A 22 interferon stimulated gene (ISG) diagonal linear discriminate analysis (DLDA) classification algorithm in a post-hoc analysis was found to be inversely ($R = -0.6$, $p = 0.04$) correlated with viral replication and tumor microenvironment remodeling including proinflammatory changes and CD8 + T cell infiltration in post treatment samples. This data supports that oncolytic MV derivatives warrant further clinical investigation and that an ISG-based DLDA algorithm can provide the basis for treatment personalization.

Glioblastoma (GBM) is the most common and aggressive malignant tumor in the central nervous system with an incidence of approximately three cases per 100,000 population in the United States (US)[1,2]. GBM patients have a dismal prognosis with a scarcity of active agents. With a median survival from diagnosis of 14–18 months[3–5] and median survival from recurrence of only 6–8 months[6–9], there is an urgent need for novel therapies for this disease.

[1]Department of Oncology, Division of Medical Oncology, Mayo Clinic, Rochester, MN, USA. [2]Department of Molecular Medicine, Mayo Clinic, Rochester, MN, USA. [3]Quantitative Health Sciences, Mayo Clinic, Rochester, MN, USA. [4]Department of Neurology, Division of Neuro-Oncology, Mayo Clinic, Rochester, MN, USA. [5]Department of Radiology, Mayo Clinic, Rochester, MN, USA. [6]Department of Neurologic Surgery, Mayo Clinic, Rochester, MN, USA. [7]Department of Laboratory Medicine and Pathology, Mayo Clinic, Rochester, MN, USA. [8]Deceased: Ileana Aderca. ✉e-mail: galanis.evanthia@mayo.edu

Despite initial promise, immunotherapy efforts in GBM treatment have failed to demonstrate consistent clinical activity to date with several negative phase III trials including ACT-IV, Checkmate-143, Checkmate-498, and CheckMate-548[10–13]. Oncolytic viruses as a novel immunotherapy treatment modality are gaining interest in the clinical community due to promising level of antitumor activity and the observation that they are not subject to the same resistance mechanisms that limit the use of chemotherapy and targeted agents[14,15]. Attenuated measles virus strains represent attractive options as oncolytic agents as evidence suggests they carry minimal safety risk to the patient and population[16,17]. MV-based virotherapy offers tumor selectivity, a potent bystander-killing effect, amenability to genetic engineering and retargeting, and excellent safety[16,18]. Strains of the Edmonston measles virus (MV) vaccine lineage have shown significant antitumor effects in pre-clinical patient derived GBM models[19–21]. In phase 1 clinical trials of patients with recurrent or refractory ovarian cancer or multiple myeloma, administration of oncolytic MV strains engineered to express either the carcinoembryonic antigen (MV-CEA) or the sodium iodide symporter (MV-NIS) was associated with the development of tumor-specific immune responses and significant anti-tumor effects;[22,23] treatment was well tolerated. Clinical trials are currently ongoing across a range of solid tumors[18].

Enhanced efficacy of virotherapy may be achievable using a pharmacogenomics approach to identify gene variants and expression signatures to preselect patients who are likely to respond to treatment. Preliminary data of a phase 1 study analyzing predictors for replication of oncolytic MV showed that constitutive activation of the interferon (IFN) pathway was a key determinant for MV replication, resulting in reduced infection of patient-derived GBM xenografts[24]. Using a diagonal linear discriminant analysis (DLDA) algorithm to predict permissiveness to viral replication, we observed that inhibition of JAK1/2, a critical component of IFN-stimulated gene (ISG) signaling, sensitized virus-resistant cells to MV infection. In ten consecutive patients with GBM who were treated by stereotactic injection of MV-CEA, elevated ISG expression was inversely correlated with MV replication[24].

Herein, we present the final analysis of a phase 1 trial (NCT00390299) that evaluated the maximum tolerated dose (MTD), safety and toxicity, as well as the preliminary efficacy of MV-CEA when administered intratumorally and into the resection cavity of patients with recurrent GBM. In addition, a 790-gene custom Nanostring panel based on microdissected tumor specimens at baseline and following one treatment dose was used to investigate tumor molecular signatures predictive of viral replication and the effect of viral treatment on tumor expression profiling of patients with GBM treated with MV-CEA. Here we show that treatment with repeat intratumoral administration of MV-CEA is safe without dose-limiting toxicity up to the maximum feasible dose and it results in proinflammatory tumor remodeling. An ISG-based DLDA algorithm predicts viral replication and can provide the basis for treatment personalization.

## Results

### Baseline demographics

In total, 23 patients were enrolled: 10 patients (9 evaluable, 1 patient withdrew prior to receiving treatment) in Group A and 13 patients in Group B. One patient enrolled in Group A did not receive study treatment, so was unevaluable. The 13 patients in Group B were enrolled following the determination of the MTD in Group A. The baseline demographics and clinical characteristics are summarized in Table 1. The median age was 53.5 years (range 37–69), 86% patients had an ECOG performance status of 0 or 1, and 91% had received ≤2 previous chemotherapy regimens; bevacizumab was received prior to the trial in 23% of patients. IDH and MGMT status for group A and B patients are included in Supplementary Table 1.

**Table 1 | Patient demographics and baseline characteristics of treated patients**

| Characteristic | Patients (n = 22) |
|---|---|
| Age, median (min, max), years | 53.5 (37.0, 69.0) |
| **Gender** | |
| Female | 11 (50.0) |
| Male | 11 (50.0) |
| **ECOG performance status score, n (%)** | |
| 0 | 6 (27.3) |
| 1 | 13 (59.1) |
| 2 | 3 (13.6) |
| **Corticosteroid therapy at study entry, n (%)** | |
| Yes | 12 (54.5) |
| No | 10 (45.5) |
| **Prior chemotherapy regimens, n (%)** | |
| 1 | 13 (59.1) |
| 2 | 7 (31.8) |
| 3 | 2 (9.1) |
| **Prior bevacizumab treatment** | |
| Yes | 5 (22.7) |
| No | 17 (77.3) |

### Safety

No DLTs were observed at any of the three dose levels for Group A and the MTD for MV-CEA was determined to be $10^7$ TCID50. Subsequently, 13 patients were enrolled in Group B, with 10 patients dosed at the MTD. No DLTs were observed for any patient and $2 \times 10^7$ TCID50 was established as the maximum tolerated Group B MV-CEA dose. In total, 14 patients (63.6%; Group A: 7 patients [77.7%]; Group B: 7 patients [53.8%]) reported a treatment-related adverse event (TRAE). Overall, 4 patients reported a Grade 2 TRAE: fatigue was reported by 2 patients [(1 patient each in Group A ($10^7$ TCID50) and B ($2 \times 10^6$ TCID50)], 1 patient had anemia [Group A ($10^7$ TCID50)] and 1 patient [Group B ($2 \times 10^6$ TCID50)] reported both lymphopenia and speech impairment (Fig. 1). There were no occurrences of any Grade ≥3 TRAEs (Supplementary Fig. 1A, B and Supplementary Table 2).

### CEA levels

There was a slight increase in CEA levels from pre- to post treatment: 0.127 ng/ml (95% CI: −0.093, 0.348), but this did not reach statistical significance (paired $t$ test $P = 0.244$). CEA elevation in the peripheral blood following treatment above the 3 ng/ml upper limit of normal was observed in one patient treated with $2 \times 10^7$ TCID50.

### Peripheral blood CD4, CD8, and complement immunoglobulin levels

No significant difference as compared to baseline was observed following study treatment.

### Assessment of viral biodistribution and shedding

There was no evidence of shedding as tested by qRT-PCR in mouth gargle and urine specimens for any of the study patients at the pre-specified time points, and no detection of viral genomes in peripheral blood.

### Assessment of immune response to MV

Figure 2 depicts mean serum anti-measles antibody levels in the serum at baseline and post treatment (4 weeks from study entry) according to patient group. As per study eligibility, all patients were measles-immune at baseline. There was no significant change in the measles antibody titers in blood during the course of the trial, as compared

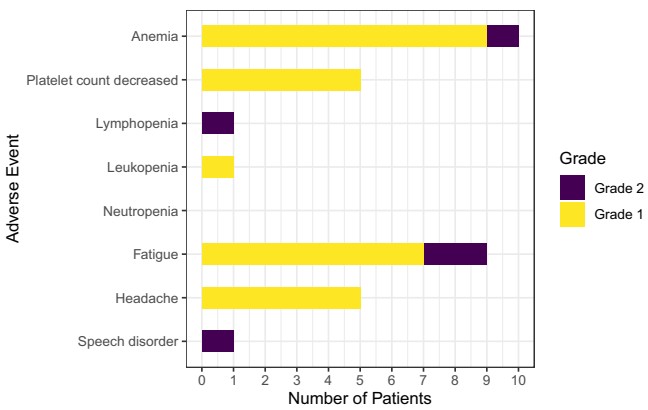

**Fig. 1 | Treatment-related adverse events.** Treatment was well tolerated with only grade 1 and 2 toxicity being observed ($n = 22$ evaluable patients).

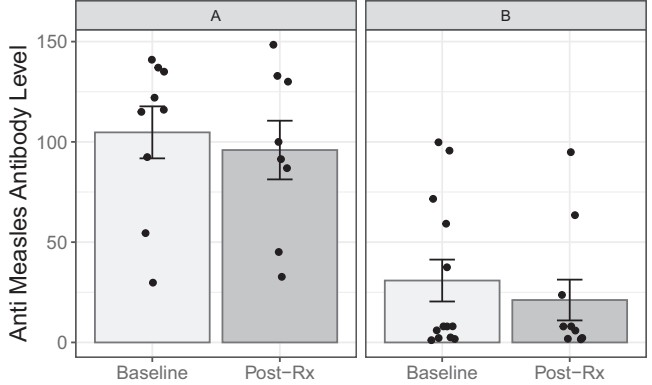

**Fig. 2 | Baseline and posttreatment anti-measles virus (MV) antibody levels.** Error bars represent the standard error of the mean (SEM) **A** pre-treatment: $n = 9$ patients, mean =104.7 +/−13.0 SEM; **A** post treatment: $n = 8$ patients, mean=95.9 +/−14.6 SEM; **B** pre-treatment: $n = 13$ patients, mean=30.9 +/−10.4 SEM; **B** post treatment: $n = 10$ patients, mean=21.2 +/−10.2 SEM). Source data are provided as Source Data file.

with baseline. In arm A there was a mean change of −6.7 units (95% CI: −17.2 to 3.9; paired $t$ test $P$ value = 0.178). In arm B, the difference was −2.7 units (95% CI: −6.1 to 0.76; paired $t$ test $P$ value = 0.112).

### Molecular profiling of tumors in MV-treated patients

Based on analysis performed on 35 patient-derived GBM xenografts and clinical patient samples we have developed a weighted gene signature, diagonal linear discriminate analysis (DLDA) classification algorithm, comprised of 22 interferon-stimulated genes (ISG). We have previously demonstrated that baseline levels of IFN response activation in the tumor were inversely correlated with virus replication[24] and we now present updated results on all 13 patients of Group B (Supplementary Data 1). Updated results confirm the initial observations; the baseline ISG DLDA score was strongly ($P = 0.04$) and inversely ($R = −0.6$) correlated with MV replication and predictive of viral infection in the patient tumors (Table 2 and Fig. 3); this analysis was limited to Group B patients because in this group tumors were resected five days following the first viral administration, which allowed correlation of the DLDA score at baseline with viral replication post treatment in resected tumor material. Because the DLDA score was developed during the conduct of the trial, the analysis was not prespecified in the clinical protocol and as such is viewed as exploratory. As shown in Fig. 3 and Table 2, patients with elevated baseline ISG

**Table 2 | Measles virus detection in tumors by qRT-PCR on day 5 following viral administration**

| Patient | Dose TCID$_{50}$ | Genome copies/µg RNA |
| --- | --- | --- |
| Pt 1 | $2 \times 10^6$ | $6 \times 10^7$ |
| Pt 2 | $2 \times 10^6$ | Not available |
| Pt 3 | $2 \times 10^6$ | $3.8 \times 10^4$ |
| Pt 4 | $2 \times 10^7$ | $4.1 \times 10^4$ |
| Pt 5 | $2 \times 10^7$ | $1.3 \times 10^5$ |
| Pt 6 | $2 \times 10^7$ | $1.7 \times 10^5$ |
| Pt 7 | $2 \times 10^7$ | $1.3 \times 10^3$ |
| Pt 8 | $2 \times 10^7$ | $6.8 \times 10^6$ |
| Pt 9 | $2 \times 10^7$ | $1.2 \times 10^4$ |
| Pt 10 | $2 \times 10^7$ | 0 |
| Pt 11 | $2 \times 10^7$ | $2.6 \times 10^3$ |
| Pt 12 | $2 \times 10^7$ | $1.6 \times 10^3$ |
| Pt 13 | $2 \times 10^7$ | $1.2 \times 10^4$ |

expression, as reflected in a high DLDA score, had significantly lower levels of virus replication. Corticosteroid use at baseline did not impact viral replication ($P = 0.876$, Supplementary Fig. 2A). In contrast, expression levels of the three known MV entry receptors, CD46, SLAM, and Nectin-4, were comparable among Group B patients (Fig. 4) indicating that the observed differences in viral replication did not result from differences in viral entry.

To identify gene expression changes following MV therapy, we analyzed post treatment (day 5) tumor samples in all Group B patients with the same 790-gene custom-made NanoString panel and performed hierarchical clustering and gene enrichment analysis. We selected genes, whose expression was at least twofold different (up- or downregulated) with a $P$ value 0.05 or less in the group of replication permissive tumors versus their expression in the group of resistant tumors. We then performed hierarchical clustering of all samples on expression values of this set of genes. We observed that samples from patients with the most permissive tumors formed one cluster (left side of heat map in Fig. 5A, underlined with the black bar) while samples from patients with the least permissive (resistant) tumors formed a cluster on the opposite branch of the hierarchical tree (right side of heat map in Fig. 5A, underlined with the magenta bar). Samples from patients with intermediate permissiveness formed clusters were located between these two extreme groups (Fig. 5A, underlined with the green bar). Next, we performed gene enrichment analysis of this set of genes, which demonstrated that multiple biological processes involved in the immune response are differentially enriched. The biological processes associated with the 14 smallest (most significant) adjusted p values for the enrichment are plotted alongside the heat map in Fig. 5B: as Fig. 5A, B illustrate, differences in viral permissiveness result in differential expression of genes associated with immune and inflammatory responses, chemotaxis and chemokine mediated signaling in posttreatment tumor samples.

### Assessment of immune cell subpopulations in tumor specimens

The pre- and posttreatment gene expression and pathway analysis results indicate that differences in viral permissiveness translate in differences in expression levels of genes mediating immune and inflammatory responses. We then sought to evaluate if this would translate in differences in the development of posttreatment immune infiltrates and tumor microenvironment remodeling. Eleven out of 13 patients in Group B had matched pre- and posttreatment (day 5) samples available for IHC analysis of immune infiltrates. There was a significant increase in CD8+ and CD68+ T-cell infiltration from baseline tumor samples to day 5 of treatment (CD8: Wilcoxon signed-rank

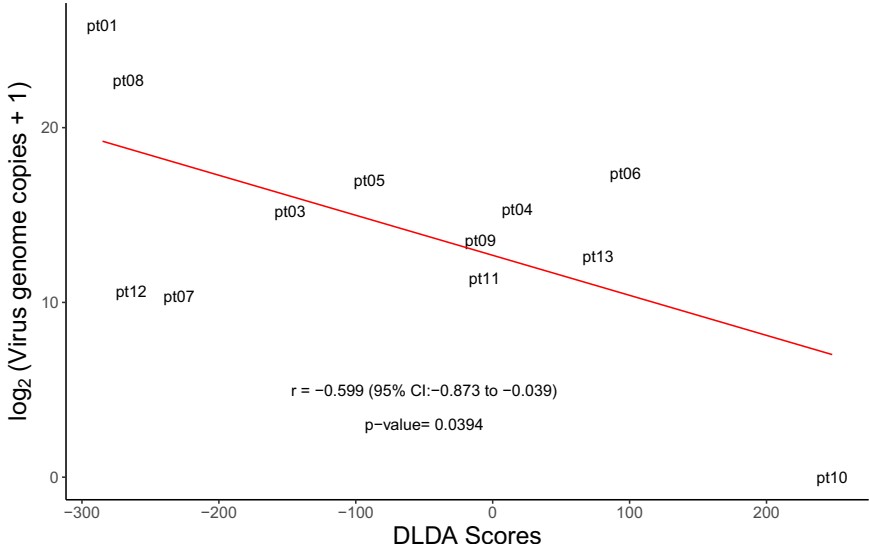

**Fig. 3 | DLDA score correlation with viral replication in treated tumors.** DLDA scores calculated for all Group B patients were correlated with virus replication in the treated tumors (*n* = 12 patients with available virus replication data). Pearson's correlation coefficient (*r*), 95% CI and two-sided *P* value based on a t-distribution with *n*-2 degrees of freedom. Source data are provided as Source Data file.

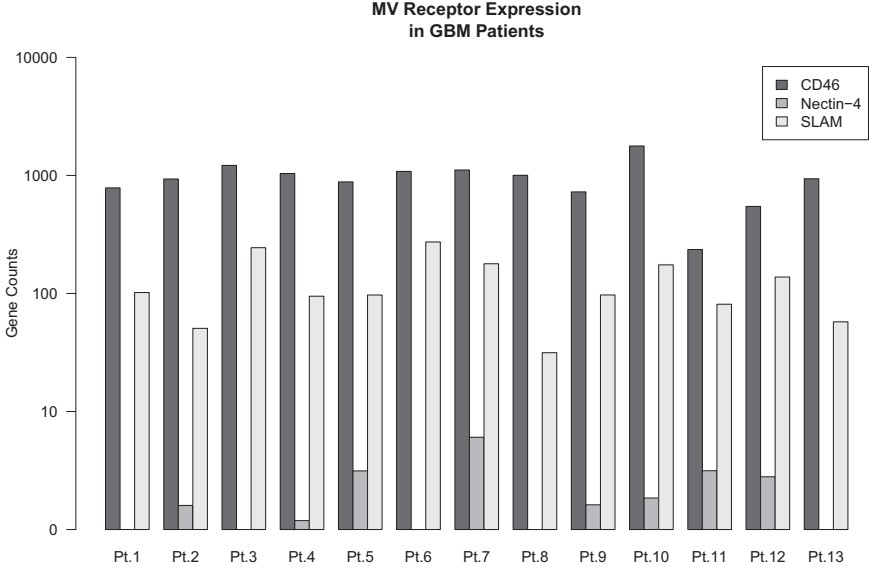

**Fig. 4 | Measles virus (MV) receptor expression in study patients.** Gene expression levels of the three known MV receptors CD46, SLAM and Nectin-4 were assessed in Group B patients (*n* = 13) and found to be comparable. Source data are provided as Source Data file.

test V = 57, *P* = 0.032, *n* = 11, Fig. 6A, B, CD68: Wilcoxon signed-rank test V = 56, *P* = 0.042, *n* = 11), but not of CD4+ cells (Supplementary Fig. 3A). There was a moderate negative correlation between lower DLDA score (i.e., increased permissiveness to viral replication) and a greater increase in CD4+ and CD8 + T cells following treatment (CD4: Spearman's rho = −0.52, S = 334, *P* = 0.11; CD8: Spearman's rho = -0.46, S = 322, *P* = 0.154, Fig. 6C and Supplementary Fig. 3B). A greater change in posttreatment CD4 +, CD8+ and CD20+ cell percentage was observed in patients who were not on corticosteroids at study entry (Wilcoxon rank-sum test CD4+ *P* = 0.042, CD8+ *P* = 0.024, CD20+ *P* = 0.042; Supplementary Fig. 2B).

Five of the 11 patients had additional surgeries performed at 1.3, 1.8, 4, 7.3, and 8 mo from study entry In 4 of 5 patients, lymphocytic infiltration had returned at or below baseline levels at the time of subsequent surgery, while in one patient the lymphocytic infiltration increased over time (Fig. 6D).

## Efficacy

Comparable median PFS was observed between the two treatment arms: Group A, 3.0 months (95% confidence intervals [CI]: 3.0–NA); Group B, 3.4 months (95% CI: 2.3, NA; HR: 0.97 (95% CI: 0.41, 2.3), *P* = 0.95). Median PFS for all study patients was 3.4 mo (95% CI: 2.9, 4.9) (Fig. 7A). Median OS was also similar between the two arms: Group A, 11.8 months (95% CI: 4.4; NA); Group B, 11.4 (4.3; NA); HR = 1.66 (95% CI: 0.67; 4.11), *P* = 0.28 (Fig. 7B). Median OS for all study patients was 11.6 mo (95% CI: 6.4; 17.8). Overall PFS rates at 3 and 6 months were 59.1% (95% CI: 41.7%, 83.7%) and 22.7% (95% CI: 10.5%, 49.1%), respectively. Overall OS rates at 6 and 12 months were 68.2% (95% CI: 51.3, 90.7) and 45.5% (95% CI: 28.8%, 71.8%), respectively. Only two study patients (both in group A) were IDH mutant. Median PFS in the study was not significantly impacted by IDH status: IDH wild-type patients had a median PFS of 3.4 months (95% CI: 3.0, 5.0) versus 4.8 months in IDH mutant patients (95% CI: 3.0, NA; log-rank *P* = 0.650). As expected, IDH

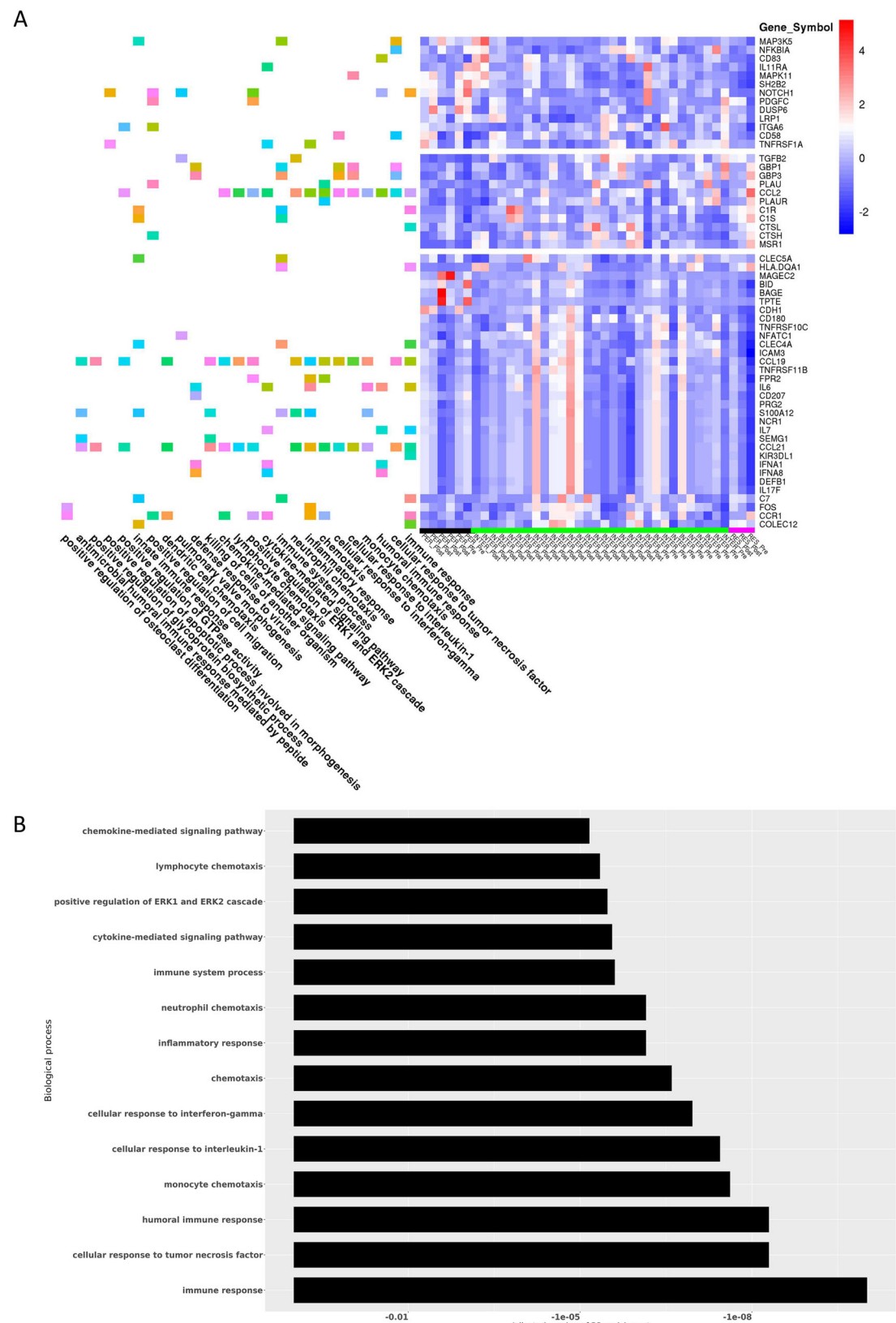

Fig. 5 | Differential gene expression in posttreatment samples of viral replication resistant versus permissive tumors (*N* = 13 patients). A RNA was isolated from tumor biopsies before and after MV therapy, expression was analyzed by Nanostring and differentially expressed genes were used to determine differentially activated pathways. The heatmap shows gene expression intensity across samples (right panel) and biological processes enriched by each gene (left panel). Permissive tumors (*N* = 3 patients) are underlined with a black bar, intermediate permissiveness (*N* = 9 patients) with a green bar and resistant tumors (*N* = 1 patient) with a magenta bar at the bottom of the heat map. Expression of the genes presented in this plot was at least twofold different (up- or downregulated) with *P* value 0.05 or lower in the group of good responders vs poor responders after the treatment. B Bar plot depiction of adjusted *P* values corresponding to biological processes (BP) of the Gene Ontology (http://geneontology.org/): The 14 processes that are most differentially enriched post treatment in patients with replication permissive versus resistant tumors are shown here.

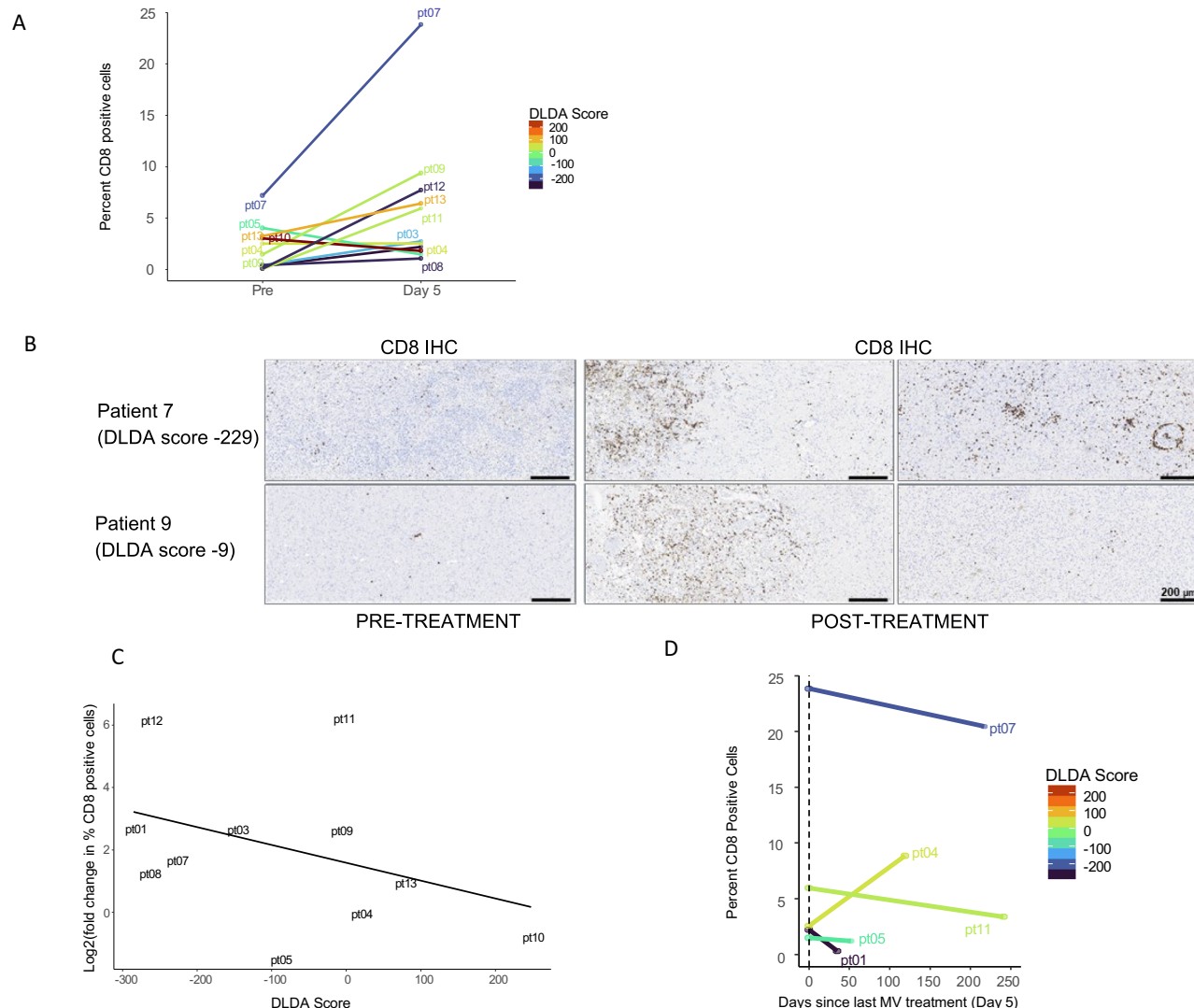

**Fig. 6 | Pre- versus posttreatment CD8 + T-cell infiltration and correlation with baseline DLDA scores in group B patients. A** There was a significant increase in CD8 + T-cell infiltration from baseline tumor samples to day 5 of treatment (CD8: Wilcoxon signed-rank test V = 57, P = 0.032, n = 11). **B** Development of CD8 + T-cell predominant lymphocytic infiltrates was observed in study patients following one dose of MV treatment, including patients with intermediate DLDA scores. Representative examples of two study patients are shown. **C** Correlation between DLDA score and CD8 + T-cell increase following treatment. Spearman's rho = −0.46 (95% CI: −0.83 to 0.19, two-sided P value, P = 0.154, n = 11 patients with available log2 fold change CD8 data). **D** Lymphocytic cell infiltration was evaluated in tumor specimens obtained in subsequent surgeries following study completion in a subset of study patients; evolution of the percentage of CD8-positive cells is depicted. Source data are provided as Source Data file.

mutant patients lived longer as compared to IDH wild-type patients: IDH wild-type patients had a median overall survival of 11 months (95% CI: 4.4, 16) versus 30 months in IDH mutant patients (95% CI: 22, NA; log-rank P = 0.031) (Supplementary Fig. 4A, B). All study patients had surgery as per trial design: the best objective response was stable disease and observed in 8 (88.9%) and 12 (92.3%) of the patients in Groups A and B, respectively. One patient in each arm had progressive disease. Supplementary Fig. 5 includes a characteristic example in a Group B patient, highlighting the evolution of imaging changes post treatment.

## Discussion

This manuscript reports on first-in-human testing of an engineered oncolytic measles virus strain administered intratumorally in the CNS for the treatment of recurrent GBM. We evaluated two different dosing strategies of this oncolytic MV strain expressing the human carcinoembryonic antigen (MV-CEA), with the aim of determining the MTD and assessing the safety and preliminary efficacy of this agent. In group

A, patients had resection of their recurrent tumor followed by virus administration into the resection cavity; the MTD was established as $10^7$ TCID50. Patients in group B first received an intratumoral injection of the MV-CEA followed by resection 5 days later; the MTD was determined to be $2 \times 10^7$ TCID50, which represented the maximum planned dose given viral titers and limitations regarding feasible CNS injection volume.

Both strategies were safe with no DLT observed in either group at levels up to $2 \times 10^7$ TCID50. There were no reports of Grade ≥3 toxicity, while 4/22 patients (18%) had grade 2 AEs at least possibly related to the virus. Immunosuppression has been observed following wild-type MV infection and can be associated with suppression of delayed-type hypersensitivity response (DTH), bacterial infections, and reactivation of tuberculosis[25]. It is, however, infrequent and transient following measles vaccination[26]. In this study, no treatment-induced immunosuppression as assessed by DTH, CD4, CD8, immunoglobulin, and complement levels was observed, an important consideration, given the immunosuppression frequently observed in glioblastoma patients,

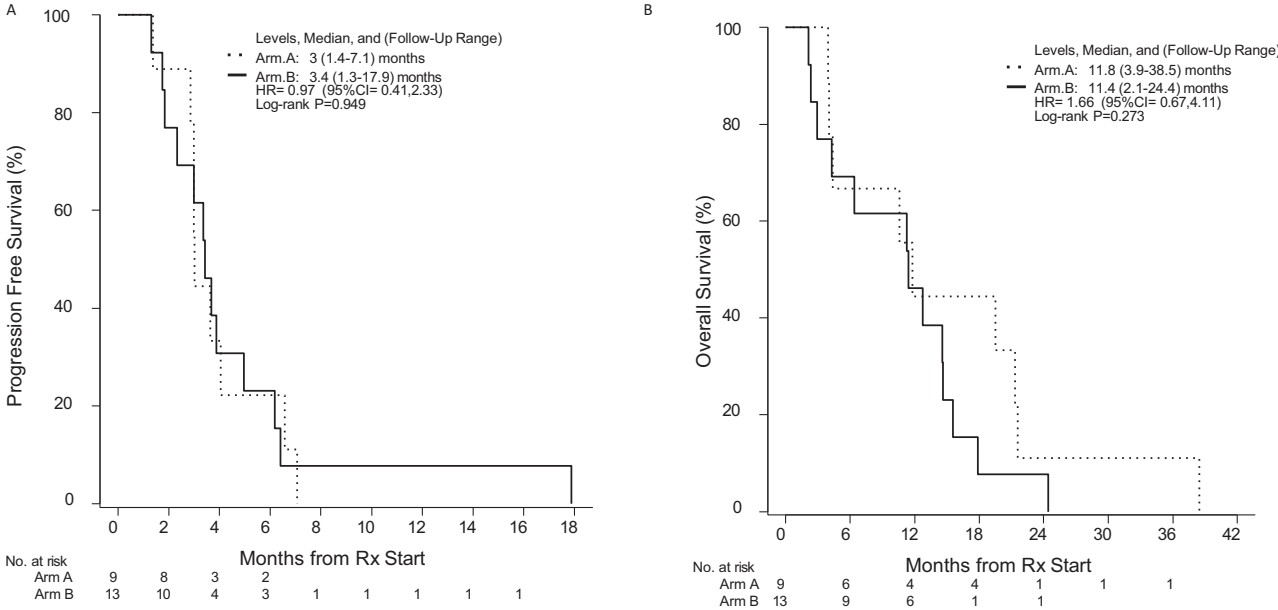

**Fig. 7 | Progression-free survival and overall survival outcomes in study patients.** **A** Progression-free survival in Group A and Group B patients. **B** Overall survival in Group A and Group B patients. Source data are provided as Source Data file.

both due to their disease and treatment[27]. Moreover, no viremia or virus shedding was observed, which confirms that MV strains exhibit high level of environmental safety.

Although efficacy was a secondary endpoint in the trial, preliminary evidence indicates that intratumoral injection with MV-CEA alone is associated with biologic activity despite the fact that 41% of the study patients had received more than 1 chemotherapy regimens and 23% had failed bevacizumab. Stable disease was observed in 88% of patients in Group A and 92.3% of patients in Group B, respectively. Observed median OS was 11.5 mo; these outcomes are favorable versus other contemporary studies in the recurrent GBM population. For example, in the contemporary Alliance A071101 trial patients with recurrent GBM at first recurrence were treated with gross total resection followed by two different combinations of heat shock protein vaccine with bevacizumab versus bevacizumab: a median survival of 6–8.5 mo was observed[28]. Other contemporary randomized data demonstrate that in bevacizumab pretreated patients, the expected median OS is dismal and in the range of 3–4 mo[29]. In addition, the OS12 rate compares favorably with the 25% OS12 observed in contemporary recurrent GBM trials, including patients at first recurrence[30,31]. Although these findings cannot be viewed as definitive given the small sample size in the trial, they support prospective validation of measles-based immunovirotherapy strategies in this patient population.

Analysis of posttreatment samples, obtained on day 5 after viral administration, confirmed detection of viral genomes to an extent that depended on baseline expression of interferon-stimulated genes in these tumors. We have previously developed a predictive algorithm, a weighted gene signature (DLDA score) based on 22 interferon-stimulated genes, that can predict viral permissiveness following MV infection in patients with GBM and ovarian cancer[24]. Applying this algorithm in all 13 Group B patients, we found that the DLDA score was also predictive of MV replication. Those patients who had lower ISG scores tended to have tumors that were more permissive to the virus and had higher levels of virus replication. Thus, patients who are more likely to potentially benefit from the MV therapy can be pre-selected by assessing the level of ISG expression. Of importance, there was a strong trend supporting a reverse correlation between DLDA score and CD8 lymphocytic infiltration in posttreatment samples suggesting that

viral replication plays an important role in the observed posttreatment remodeling of tumor microenvironment.

It is of note that 2 of the 13 patients in group B (15%) had a low DLDA score and were very permissive to viral replication despite the fact that one of these patients was treated in the lower dose level ($2 \times 10^6$ TCID50). Given the importance of interferon response pathway in controlling the replication of most other oncolytic viruses, this data suggests that approximately one out of five GBM patients would have tumors exhibiting the degree of permissiveness that results in optimal viral replication. This hypothesis is corroborated by clinical observations with several other viruses including Delta-24-RGD[32], poliovirus[33], and Toca-511[34] where response rates ranging from 9–13% were observed. It also highlights one of the possible challenges in the field of oncolytic virotherapy in GBM: if the benefit of single-agent therapy is expected to be confined to <20% of patients who have a suppressed baseline response against interferon, phase III trials in unselected patients would have a very high likelihood of being negative by diluting the virotherapy responsive patient population[35].

Viral treatment resulted in tumor microenvironment remodeling with significant increase in CD8+ cell infiltrates on day 5 posttreatment samples. Of note, our data illustrates that those patients with moderate DLDA score ( <150 and > −250) can still accomplish intermediate levels of viral replication which is adequate to allow them to benefit from the immunostimulatory effect of even limited viral replication on tumor microenvironment; upregulation of proinflammatory mediators and lymphocytic infiltration on day 5 biopsy was observed. Viral infection was found to induce increases in proinflammatory cytokines and chemokines (Fig. 5A, B) that have previously been shown to be involved in creating an immunogenic environment[36] and stimulate immunologic response[37]. This type of localized inflammation has the potential to augment the effector functions of infiltrating immune cells, facilitate the generation of antitumor immunity, and counteract tumor-induced immunosuppression[38]. It is also possible that repeat (beyond two doses) administration of our MV oncolytic strain could have the potential to further enhance this effect and maintain the proinflammatory remodeling of the tumor microenvironment, that otherwise may reverse. Although the impact of multiple dosing cannot be addressed by our study, data generated with repeat administration

of the herpes virus strain G47Δ in Japanese patients appear to support this approach[39].

Overall, our data indicates that intratumoral administration of measles virus strains can act as a form of in situ vaccination inducing microenvironment changes that can facilitate tumor-immune recognition and help reverse resistance to other immunotherapies such as immune checkpoint inhibitors. Indeed, in parallel preclinical work in the GL261 and CT2A models, we have demonstrated that combination of oncolytic measles virus strains with murine anti-PD1 resulted in synergy with 60–80% of animals being cured[20,40]. In addition, the development of tumor-specific immunologic memory was observed as demonstrated by the fact that 100% of the surviving animals remained disease-free when rechallenged with the autologous glioblastoma lines, but not when rechallenged with different tumor (melanoma) line[40]. Thus, patients with intermediate DLDA scores, although not benefiting from single-agent virotherapy as much as patients with low DLDA scores, could still be excellent candidates for combinatorial strategies, for example, with immune checkpoint inhibitors or agents that block activation of the interferon response pathway. We have also demonstrated that even low levels of oncolytic viral replication are adequate for the synergistic effect of virotherapy with anti-PD1 inhibition to materialize, and that increasing viral replication with the use of JAK/STAT inhibitors, which block the interferon response pathway, can further enhance this effect[20,40].

Despite promising clinical data in this area, as with any new agent, further studies are required to fully assess the translation and clinical applicability of this agent, and other MV strains. In this first-in-human study in glioblastoma the Edmonston measles oncolytic platform demonstrated safety, ability to replicate in the tumor and resulted in promising survival outcomes. Despite the fact that all patients were immune to the virus per study design and FDA mandate in order to increase safety, systemic pre-existing immunity did not block replication in the tumor, following intratumoral administration. Our data also demonstrate that variability in viral permissiveness needs to be considered, emphasizing the importance of patient selection. The DLDA-weighted gene signature we have developed along those lines could help individualize treatment and allow us to select patients that may derive optimal benefit from single-agent virotherapy versus combinatorial strategies[24]. Furthermore, in order to further enhance the immunostimulatory potential of oncolytic cell death, we have engineered measles strains to express the Helicobacter pylori neutrophil-activating protein (NAP), a potent TLR2 agonist. We have demonstrated a significant increase in activity in immunocompetent GBM models, enhancement of immunostimulatory response with increased secretion of damage-associated molecular patterns such as HMBG1 and calreticulin, and synergy with immune checkpoint inhibitors[40].

In conclusion, these data of oncolytic measles virus strains in recurrent GBM patients create an important foundation that supports subsequent testing of MV strains with immunostimulatory payloads as well as strategies for treatment individualization.

## Methods
The study (https://clinicaltrials.gov/study/NCT00390299, registration date 10/19/2006) was designed and conducted in accordance with the provisions of the Declaration of Helsinki and Good Clinical Practice Guidelines. The Mayo Clinic Institutional Review Board Committee approved the protocol, which was conducted under the oversight of the Mayo Clinic Cancer Center Data and Safety Monitoring Board.

### Patients
Eligible patients were aged ≥18 years with recurrent Grade 3 or 4 glioma, including astrocytoma, oligodendroglioma or mixed glioma with histologic confirmation at initial diagnosis or recurrence, who were candidates for gross total or subtotal resection. Patients were required to have an Eastern Cooperative Oncology Group (ECOG)

performance status of 0–2, anti-measles virus immunity as demonstrated by immunoglobulin G (IgG) anti-measles antibody levels of ≥1.1 EU/mL by ELISA, normal serum CEA levels (< 3 ng/mL), and adequate hematologic, hepatic and renal function. Ineligible patients included those with an active infection ≤5 days prior to enrollment; history of tuberculosis or a positive skin test; who had received noncytotoxic antitumor drugs ≤2 weeks, or chemotherapy, immunotherapy or biologic therapy ≤4 weeks, or radiation therapy ≤6 weeks, or bevacizumab treatment ≤12 weeks prior to enrollment, or any viral or gene therapy prior to enrollment; who had failed to fully recover from acute, reversible effects of prior chemotherapy; with New York Heart Association class III or IV heart failure; with exposure to household contacts ≤15 months old or household contact with immunodeficiency, or had allergy to measles vaccine or history of severe reaction to prior measles vaccination; who were HIV-positive or history of other immunodeficiency. Written informed consent was obtained from all patients and no patient received compensation for study participation.

### Study design
This single-arm, phase I/II trial enrolled patients in two treatment regimens in a standard 3 + 3 cohort design, with additional patients enrolled in a maximum tolerated dose (MTD) expansion cohort. Cohorts of three patients were enrolled in Group A until the MTD of MV-CEA was determined following single-dose administration. Subsequently, cohorts of 3 patients were enrolled in Group B for further evaluation of the MTD following administration of two viral doses. Additional patients were then enrolled in the Group B MTD expansion cohort. The first study patient (Group A) was enrolled on 10/23/2006 and the last patient (Group B) was enrolled on 11/30/2019. The study schema is summarized in Supplementary Fig. 6, and the protocol is included as Supplementary Note in the Supplementary Information file. In Group A, patients underwent *en bloc* tumor resection on Day 1, followed by administration of MV-CEA into the resection cavity at viral tissue culture infectious doses 50% (TCID50) of $10^5$–$10^7$. The viral dose was diluted in 1 mL of saline and delivered via a 20-gauge blunt tip needle injected 1–2 cm into the brain parenchyma at ten injection sites. The starting dose of $10^5$ TCID50 (dose level 1 [DL1]) was increased to $10^6$ TCID50 (DL2) and $10^7$ TCID50 (DL3) if ≤1 dose-limiting toxicity (DLT) was observed at the previous dose level. In Group B, recurrent GBM patients had a silastic ventricular catheter placed using stereotactic equipment, with computed tomography/magnetic resonance imaging employed to secure catheter placement. On Day 1, patients received an intratumoral injection of MV-CEA at viral doses of $2 \times 10^6$–$2 \times 10^7$ TCID50, with the dose administered as a single 1 mL bolus diluted in saline and injected into the tumor at 0.1 mL per minute via the catheter. The catheter was secured to the dura and left in place to mark the injection site. On Day 5, patients underwent en bloc tumor resection before receiving a second injection of MV-CEA into the resection cavity, which was similarly diluted in 1 mL of NS and administered via a 20-gauge blunt tip needle in multiple sites of the resection cavity wall. Supportive care was given to all patients, including blood products, anticonvulsants, perioperative steroids, antibiotic therapy, and treatment of other newly diagnosed or concurrent medical conditions. The study protocol was approved by the Mayo Clinic Institutional Review Board.

### Safety and efficacy assessments
Assessment of safety and toxicity with determination of MTD was the study's primary endpoint. Secondary endpoints included progression-free survival and correlative analysis, including assessment of viral replication in tumor, viremia, and shedding. All adverse events (AEs) were evaluated per the National Cancer Institute (NCI) Common Terminology Criteria for Adverse Events (CTCAE). Patients were assessed at 4 weeks following day 1 viral administration for toxicity including DLT, then followed up every 2 months for disease

progression and survival. DLTs were defined as those AEs definitely, probably or possibly attributed to the study treatment that met the following toxicity criteria: hematologic, defined as Grade ≥3 except Grade 3 neutropenia lasting <72 h; nonhematologic, defined as Grade ≥3 (Grade ≥3 nausea, vomiting, or diarrhea was considered dose-limiting only if the patient is receiving the maximum supportive care regimen described in the protocol; alopecia will not be considered a DLT); neurologic, defined as Grade ≥2; allergic reaction, defined as Grade 2 asymptomatic bronchospasm and/or urticaria, and Grade ≥3 allergic reactions; or viremia lasting ≥6 weeks from last viral administration. The MTD was defined as the dose level below the lowest dose that induced DLT in at least two of six patients.

Patients were evaluated for treatment response at 4 weeks after tumor resection and every 2 months until disease progression. Progression-free survival (PFS) was defined as the time from registration to documentation of disease progression. Patients who died without documentation of progression were considered to have had tumor progression at the time of death unless there was documented evidence that no progression occurred before death. Response assessment was performed according to the RANO criteria[41].

## GBM patient sample preparation and analyses

Formalin-fixed paraffin-embedded tumor tissue samples collected during the primary and recurrence surgery were used for histological staining immunohistochemical (IHC) analysis and NanoString analysis (custom-modified nCounter Pan-Cancer Immune Profiling Panel). RNA isolated from fresh/frozen tumor tissue was tested with qRT-PCR in order to detect virus replication. Tumor tissue removed en bloc with the catheter tip in place was measured and photographed in our pathology laboratory (Supplementary Fig. 7). The catheter was subsequently removed and fresh tissue was prepared into serial sections 2–3 mm in thickness, perpendicular to the catheter. Sections for formalin-fixing and freezing were photographed to establish subsequent correlations between the tissue and relationship to the catheter. Frozen tissue was stored at −80 °C.

## Immunohistochemistry of pre- and posttreatment samples for assessment of immune cell subpopulations

All histological sections were reviewed, and 4-μm sections were obtained from the most representative formalin-fixed paraffin-embedded (FFPE) tissue blocks. Immunohistochemical stains (IHC) were performed utilizing antibodies directed against CD3 (clone LN10, dilution 1/250, Leica Biosystems, UK), CD4 (clone SP35, Ready to Use Predilute Antibody, Ventana, USA), CD8 (clone C8, dilution 1/250, Dako, Denmark), CD20 (clone L26, dilution 1/300, Dako, Denmark), and CD68 (clone KP1, dilution 1/1500, Dako, Denmark) utilizing clinically validated protocols. Slides were scanned at ×40 magnification on the Aperio GT450 brightfield instrument (Leica Biosystems). The resolution of the images was 0.26 μm/pixel at ×40. The images were 24-bit contiguous standard pyramid tiled TIFFs compressed via JPEG with a quality setting of 91. A board-certified neuropathologist selected and annotated regions for analysis using Aperio ImageScope Software (Leica Biosystems). The annotated regions of each stained slide were analyzed using proprietary nuclear and cytoplasmic algorithms. Cells within each region of interest were graded based on intensity of staining (0, 1+, 2+ or 3+). Cells with an intensity of 1+ or higher were considered positive for immunostaining markers. Immunohistochemistry (IHC) scores were expressed as a percentage of positive cells (0 to 100) within the region of interest.

## Peripheral immune response, CEA levels, and viremia assessment

MV-specific immunity was assessed by ELISA to measure anti-MV-specific IgG levels at baseline, 28 days after study entry and every

2 months until progression. Peripheral blood CEA levels were assessed at the Mayo Clinic Central Clinical Lab using the Bayer Diagnostics Advia Centaur Immunoassay system (Bayer Healthcare Diagnostics). Viremia and viral shedding were assessed by qRT-PCR from patient peripheral blood mononuclear cells, throat gargle specimens, and urine samples. The schedule for the correlative laboratory analysis is summarized in Supplementary Fig. 8.

## Quantitative qRT-PCR for the detection of virus replication

The qRT-PCR assay was optimized for primers, probe, and magnesium concentration with TaqMan RNA to CT 1-step kit (Thermo Scientific). A 50-μL qRT-PCR reaction volume was used to amplify the MV-N genomic RNA target, in the presence of 0.3 mmol/L each of forward (5′-GGG TGT GCC GGT TGG A-3′) and reverse (5′-AGA AGC CAG GGA GAG CTA CAG A-3′) -primers, 0.2 mmol/L Black Hole Quencher–labeled probe (5′-/56-FAM/TGG GCA GCT CTC GCA TCA CTT GC/ 3BHQ_1/-3′), 4 mmol/L MgCl, and 1 mcg or a maximum total volume of 5 mcl of the RNA isolate. One cycle of reverse transcriptase reaction (15 min at 48 °C) was applied, followed by a denaturation step (10 min at 95 °C) and 40 cycles of amplification (15 s 95 °C and 1 min 60 °C), with fluorescence measured during the extension. A standard curve of tenfold dilutions containing $10^7$ to 10 MV-N gene copies/mL was generated using a manufactured RNA oligo (IDT, San Jose, CA) Quantification and subsequent calculation of copy number was done using the standard curve and the ROCHE480 Quantitative PCR System software. Total RNA from frozen tumor tissue and primary GBM lines was extracted using the RNeasy kit (Qiagen). On the first step of the reaction, cDNA was made using 25 ng of the RNA extracts and One-Step RT-PCR master mix reagents and TaqMan Probe-Based Gene Expression assays (Life Technologies). The assays were run on a Roche 480 Light Cycler instrument (Roche). The relative quantification was performed using human eukaryotic 18 S rRNA as a reference. Expression was calculated in fold change of expression as compared to corresponding normal tissue control using the comparative Ct method[42].

## Nanostring analysis

Patient tumor samples obtained at the time of primary surgery and recurrent tumor resection were examined by a pathologist to identify regions of ≥90% tumor involvement, which were subsequently scraped, and the RNA harvested. The RNA (100 ng) was hybridized with NanoString probes according to the manufacturer's protocol and using the nCounter Pan-Cancer Immune Profiling Panel (NanoString Technologies, Seattle, WA) that was custom-modified in our laboratory with the addition of 30 gene probes (Supplementary Data 2). The resulting custom NanoString was used to determine the expression of 790 genes in individual patient tumor samples. Samples were analyzed using the nCounter Digital Analyzer. NanoString results were analyzed and values normalized to housekeeping genes.

## Diagonal linear discriminant analysis (DLDA)

The DLDA scoring system, a method of classifying prospective tumors into known categories based on gene expression signatures[43,44], was used to generate an algorithm for predicting MV permissiveness in GBM. This unique system relies on a weighted gene voting scheme to influence the classification of a prospective sample. A 22 ISG gene panel identified in an earlier pathway enrichment analysis[24] and standardized to all 790 genes of the custom NanoString was utilized as the gene signature for MV permissiveness in this study. The weighting of each gene was determined according to a training data set using the permissive cell lines, GBM43 and GBM64, and the resistant cell lines, GBM150, GBM6, and GBM39. The training set DLDA model yields coefficients and a constant that can be used with other similarly standardized expression data to predict if a sample is MV-resistant or MV-permissive. A validation data set consisted of 35

GBM patient-derived xenografts[24]. Gene expression profiles of prospective tumor samples from patients with GBM were normalized and entered into the algorithm to calculate a DLDA score. A score above 150 is associated with no detectable virus indicating MV-resistance, with scores below −250 associated with the highest level of virus recovered from the tumor (MV-permissive). Intermediate values are associated with viral permissiveness at an ~2-log lower level than that associated with higher permissiveness.

## Pathway enrichment analysis

We used GeneCodis online resource (http://genecodis.cnb.csic.es/) to perform gene/protein enrichment analysis of gene expression data obtained by NanoString analysis of tumor samples obtained pre-treatment and post treatment following one dose of the virus in group B patients[45,46]. In total, 19,835 genes were used as an input into Gene-Codis and we selected genes with changes between sample groups of at least 1.5- or 2.0-fold and $P$ value < 0.05. GeneCodis outputs a listing of functional groups (GO biological process, GO molecular function, KEGG) with gene/protein IDs assigned to each group. We used a custom script in R programming language to combine fold change values and $P$ values with a functional group of proteins[47].

## Statistical analyses

Standard cohorts of three design was applied for this phase I trial[48]. Evaluable patients were those who gave their informed consent and received MV-CEA treatment.

MTD was defined as the dose level below the lowest dose that induces dose-limiting toxicity (DLT) in at least one-third of patients graded according to NCI Common Terminology Criteria for Adverse Events (CTCAE) version 3.0. Dose-limiting toxicities include hematologic events grade 3 or higher (except grade 3 ANC lasting <72 h), nonhematologic events graded 3 or higher (except grade 3 nausea, vomiting, or diarrhea were to be considered DLT only if patient was receiving the max supportive care and alopecia was not considered dose-limiting), neurologic toxicity grade 2 or higher, grade 2 allergic reactions asymptomatic bronchospasm and/or urticarial, grade 3 or higher allergic reactions, viremia lasting for 6 weeks or more from last viral administration deemed at least possibly related to treatment.

The percentage of patients who are progression-free at 3 and 6 months (PFS3 and PFS6) was summarized descriptively. Progression-free survival was defined as the length of time from the date of registration to (a) date of progression or death due to any cause or (b) last follow-up.

Kaplan–Meier survival curves and log-rank tests were used to obtain median PFS and OS times. The distribution of PFS and OS was estimated for groups A and B separately before combining using the Kaplan–Meier method. Descriptive statistics and simple scatterplots were employed to present the CEA, CD4, CD8, MV antibody immunoglobulin data, as well as data on viremia and shedding. The relationship between viral replication and DLDA score was assessed using Pearson correlation.

For the 790-gene custom NanoString analyses, to allow analysis on a common scale the gene expression values for each sample were transformed by adding 1.0 to each gene expression value then transforming each value by $\log^2$ to a standardized mean of 0.0 and standard deviation of 1.0. Differences in gene expression among tumor samples were evaluated using unequal-variance $t$ tests. $P < 0.05$ were considered statistically significant. All statistical tests were two-sided.

Trends in immune filtration were summarized and evaluated in relation to time as well as DLDA score and were graphically assessed using line plots. Pre- vs. posttreatment marker level changes were quantified and were analyzed using paired analyses as well as fold changes for percent-positive cells from pre-treatment vs. day 5 of treatment, where a standard log2 transformation was used. Univariate modeling was used as well as scatterplots and Spearman rank

correlation tests to assess the relationships between DLDA score and both baseline marker levels and log2 fold change in percent-positive cells. Given the limited number of patients with paired sample data available, multivariable or more complex modeling was not employed.

## Reporting summary

Further information on research design is available in the Nature Portfolio Reporting Summary linked to this article.

## Data availability

All requests for raw and analyzed data will be reviewed by the Mayo Clinic (MC) Institutional Review Board (IRB). Patient-related data not included in the manuscript were generated as part of a clinical trial and are subject to patient confidentiality. Any data and materials (e.g., tissue samples or imaging data) that can be shared will need approval from the MC IRB and a material transfer agreement in place; this process requires an average of 6 months. All data shared will be de-identified and will be available for 1 year after access is granted. Any requests for clinical data should be addressed to the corresponding author Evanthia Galanis (galanis.evanthia@mayo.edu). The study protocol is available as a Supplementary Note in the Supplementary Information file. The remaining data are available within the Article, Supplementary Information or Source Data file. Source data are provided with this paper.

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

## Acknowledgements

This work was supported in part by NIH grants R01CA258239, P50CA 108961, R21CA 123839, and The Ben & Catherine Ivy Foundation. The authors would like to express their gratitude to patients and their families. They would also like to thank Mrs. Susan Steinmetz for her support in coordinating patient care, Marie R. Passow, SCT (ASCP) who in conjunction with Dr. Aditya Raghunathan optimized the digital image analysis algorithms, and Mrs. Raquel Ostby for help with manuscript preparation.

## Author contributions

Conceptualization: E.G. Data curation: E.G., S.K.A., X.W.C., and I.A. Formal analysis: E.G., S.K.A., C.B.K., X.W.C., and I.A. Funding acquisition:

E.G. Investigation: E.G., C.B.K., I.A., J.H.U., J.E.H., R.S.M., S.I.R., D.R.J., T.J.K., J.C.B., K.E.D., D.H.L., T.C.B., C.G., A.R., and I.F.P. Methodology: E.G., M.J.F., A.A.L., I.A., K.B.V., C.G., A.R., I.D.I., and I.F.P. Writing— original draft: E.G. and S.K.A. Writing—review and editing: E.G., S.K.A., C.B.K., X.W.C., J.H.U., M.J.F., A.A.L., K.B.V., J.E.H., R.S.M., S.I.R., D.R.J., T.J.K., J.C.B., K.E.D., D.H.L., T.C.B., C.G., A.R., I.D.I., and I.F.P.

## Competing interests

The authors declare no competing interests or other interests that might be perceived to influence the interpretation of the article. Outside of this submission, EG has received honoraria for advisory board participation from Kiyatec, Inc. (personal compensation) and Karyopharm Therapeutics, Inc. for Data Safety and Monitoring Board participation (compensation to the employer). Her institution has received grant funding from Servier Pharmaceuticals LLC (formerly Agios Pharmaceuticals, Inc.), Celgene, MedImmune, Inc. and Tracon Pharmaceuticals. The remaining authors declare no competing interests.
