## [Peer Review File · Nature Communications]

Carcinoembryonic Antigen-Expressing Oncolytic Measles Virus Derivative in Recurrent Glioblastoma: a Phase 1 TrialREVIEWER COMMENTS

Reviewer #1 (Remarks to the Author): with expertise in oncolytic viruses, GBM

In this first in human trial of an oncolytic measles, the authors treat 23 patients with recurrent malignant gliomas via a 2 arm study: in arm A 3 dose-escalation of the agent was done via peritumoral injection, while in arm B an initial injection of the measles was done followed 5 days later by resection and peritumoral injection. The authors show safety of the agent. In addition, they provide important evidence of a 22 interferon gene signature profile inversely correlated with MV replication and infection in the arm B subjects where post-injection tumors were available. Additional, nanostring analyses clustered permissive and non-permissive tumors based on pathways associated with immune responses. In addition, there was also a correlation between increased tumor permissiveness to the MV and greater TILs in tumor.

These findings are highly significant in that they correlate MV replication with increased TILs in a tumor such as malignant glioma, known to be lymphocyte-depleted. Therefore, they authors convincingly demonstrate that MV via its replication can alter the tumor microenvironment and enrich it in TILs.

I have some minor comments primarily as a means of clarification and not as criticisms for this otherwise highly significant paper:

- 1- Patients were on corticosteroids. Was there any effect of dose or time on steroids on tumor permissiveness or induction of TILs?
- 2- Table 2- what is the genome copies/ ug RNA of the input dose? I am trying to determine if there is an increase, decrease or no change in genome copies 5 days after injection.
- 3- Figure 2- Since there was no CEA detected peripherally and MV antibodies did not rise, it seems that there is no spillage of the agent systemically after tumor injection. This provides further evidence of safety.
- 4- Figure 3- was there any correlation between the DLDA scores, MV receptor expression or the Nanostring analyses and survival?
- 5- Figure 4- can the authors divide up subject PFS and OS based on IDH mutation presence, 1p/19q deletion?

Reviewer #2 (Remarks to the Author): with expertise in oncolytic viruses, GBM

Remarks to the Author:

Galanis et al. report the results of the FIH trial of MV-CEA in 22 patients with recurrent glioblastoma. The MV-CEA virus was constructed by introducing the extracellular N-terminal domain of human CEA in the measles virus backbone derived from the Edmonston vaccine lineage. The authors evaluate two different dosing strategies of the MV-CEA to assess the safety and preliminary efficacy. They also used the DLDA scoring system to predict MV permissiveness in GBM. The authors stated that patients with elevated baseline ISG expression, as reflected in a high DLDA score, had significantly lower levels of virus replication: a score above 150 indicates MV resistance, with scores below -250 associated with MV-permissive. An ISG-based DLDA algorithm could provide the basis for treatment personalization. This concept might help select patients who could benefit from MV-CEA

treatment.

1- It would be helpful to the readers to show serial magnetic resonance images (MRI) to assess the therapeutic effect of MV-CEA during and after intratumoral administration. For example, although the patients in Group B were candidates for gross total or subtotal resection, pre- and post-resection MRI would illustrate pre-dose tumor size, the extent of resection, and changes in contrast lesions or FLAIR high-signal regions associated with anti-tumor immune response.

2- Authors have previously shown that viral propagation correlates with CEA levels, which means they could monitor the activity of the virus by testing CEA levels in the peripheral blood. However, only one patient had an elevated MV-CEA in the peripheral blood after administration in this study. Please provide pre- and post-treatment CEA levels for all cases, as this will explain that CEA levels may have increased partially.

3- The trial title states that it is for recurrent glioblastoma. However, the selection criteria are recurrent Grade 3 or 4 gliomas, including astrocytoma, oligodendroglioma, or mixed glioma with histologic confirmation at initial diagnosis or recurrence. IDH mutations are associated with better prognosis than IDH wild-type tumors of the same WHO grade (e.g., see <https://pubmed.ncbi.nlm.nih.gov/32557428/>). Initial diagnosis and IDH status should be included in supplemental tables.

4- The authors also asserted that there was a moderate negative correlation between lower DLDA scores and a greater increase in CD4+ and CD8+ T cells following treatment. However, it is unclear whether CD4+ T cells increased after treatment, as the results were not shown. Please illustrate pre- versus post-treatment CD4+ T cell positivity and correlation with baseline DLDA scores in Group B patients. It would be more convincing if the authors could present representative pre- and post-treatment CD4+ and CD8+ T cell positivity of patients with scores above 150, below -250, and intermediate values (< 150 and > -250).

5- The protocol states that adverse events should be graded at each evaluation, and pre-treatment symptoms and conditions should be evaluated at baseline. It is appropriate that Figure 1 illustrates treatment-related adverse events. However, since this is an FIH study of MV-CEA and patients will undergo en bloc tumor resection at the time of administration, unrelated adverse events occurring after MV-CEA administration should also be shown in a supplementary table.

6- Discussion - The OS of patients with resectable recurrent GBM is relatively good. The details of the A071101 trial indicate that the OS is around 8 months for patients using HSPPC-96, but among patients treated per protocol, the OS for bevacizumab alone is 12.3 months. The authors states observed median OS of 11.5 months compares favorably with other contemporary studies in the recurrent GBM population. However, since the method of administration to demonstrate efficacy will be studied in the future, it is better not to emphasize efficacy here.

7- Statistical analyses – progression-free survival was defined as the length of time from date of registration to a) date of progression or death due to any cause or b) last follow-up. However, on page 6, PFS was defined as the time from start of study therapy to documentation of disease progression. Which is correct? How long did it take from the registration date to the first dose of MV-CEA? The starting point of the survival curve would usually be the surgery date or the start of study therapy.

8- Figure 2 – Comparison within each group shows no difference in antibody levels before and after treatment, but there is a significant difference between Group A and Group B. Please provide evidence that there is no problem adapting 107 TCID50 determined to be MTD from Group A to Group B.

9- Figure 3- Figure 3E shows a moderate negative correlation between lower DLDA scores and a greater increase in CD8+ T cells after treatment. However, adding patient numbers to the black dots in the bottom graph would be clearer to clarify the relationship with the patients in the top graph. The time point indicated by the pre-treatment baseline in Figure 3F needs to be clarified. The values where the dotted line intersects the vertical axis differ significantly from the pre-treatment values in Figure 3E. For example, the percentage of CD8-positive cells before treatment for pt 07 is about 5% in Figure 3E, while the vertical axis in Figure 3F crossed with the dotted line is 20%. Please correct Figure 3F.

10- Figure 4-Since DLDA scores and survival time for each patient are not shown in tables, it is difficult to know to what extent the DLDA scores affected survival in Figure 4. Please include in tables.

Reviewer #3 (Remarks to the Author): with expertise in oncolytic viruses, GBM

This is an important clinical study from a team that has championed oncolytic virotherapy for glioblastoma. The authors have completed yet another human study which in and of itself is an incredible accomplishment. I find the work very important, solid, and in need of publication and dissemination. My remarks below are for consideration.

1. Table 1. Could we gain some more information about IDH status or MGMT status of these patients?
2. Table 2: Measles virus is detected by qPCR....do the authors have any data to support that these are viable and fully assembled viral particles? This would be helpful in the overall discussion.
3. Figure 2: I find this somewhat interesting since we usually see antiviral immunity limit OV. It would appear that anti-MV antibody levels go down in both arms, though probably not statistically significant...how do the authors explain this phenomenon?
4. Digging a little deeper...was the survival affected at all by IDH, MGMT expression?
5. Does the level of viral replication in patients correlate in any way with CD46 or nectin or SLAMF?
6. Figure 3C...the differential gene expression is interesting...have the authors validated any of these targets at the protein level?
7. Besides CD8 analysis...is there are information of CD4 levels or Tregs?
8. Statement in the discussion suggests that MV-CEA is associated with biological activity despite the fact that 41% of patients had received chemo...can the authors expand on their thinking here? I am not sure what the message is and this is probably more me than them.

Reviewer #4 (Remarks to the Author): with expertise in biostatistics, bioinformatics

The authors published a JNCI paper in 2018 about the safety and efficacy of a novel

oncolytic Measles virus (MV) derivative in patients with glioblastoma based on a preliminary data of their ongoing clinical trial. In addition, they built a 22-gene signature for predicting MV permissiveness and validated it on subjects participated up to that time.

Based on the final analysis dataset of the clinical trial with three additional subjects, the current manuscript presents an updated result of safety/efficacy of the MV derivative as well as a validation result of the 22-gene signature.

The result showed that the MV therapy is safe with no severe adverse events and has a comparable efficacy to contemporary treatments for glioblastoma. There was a significant increase in CD8+ and CD68+ T cell infiltration in 11 patients' tumors. The 22-gene signature score has a little weakened but still significant negative correlation ($r=-0.6$, $p=0.04$) to the degree of MV replication.

Overall, the work is well executed and statistically sound. I have minor comments as follows:

1. Regarding Study Design section on page 4 and 5, it appears that patient group A and B are from the 3+3 design cohort or the dose expansion cohort, respectively. However, the group were not defined in the study design section. Please clarify it in the manuscript and the trial schema (the supplementary Figure 1).

2. In DLDA section on page 8, two cutoffs of -250 and 150 for DLDA score were used to classify MV resistance. However, there is no description about how those cutoffs were derived.

3. Regarding Figure 1 on page 10, trial participants were treated at different doses of the MV therapy. Were there any difference in AEs between dose levels? I suggest stratifying them by dose levels and present or compare adverse events accordingly.

4. In Molecular profiling of tumors in MV-treated patients section on page 11, DLDA score has a significant negative correlation, but it largely depends on a few extreme observations such as Pt1 and Pt10. Please provide a 95% interval of the correlation coefficient. It is also suggested to make 3 groups of subjects by using the two DLDA cutoffs (< -250 , intermediate, > 150) and then compare the degree of virus replication. This could be another way to validate the DLDA score and justify the relevance of DLDA cutoffs.

5. On page 12, it is hard to follow that "We observed ... permissive tumors formed one cluster (right side of heat map in Figure 3C) resistant formed a cluster on the opposite branch of the hierarchical tree (left side of heat map in Figure 3C)". Please label samples or add marks in Figure 3 that will help distinguish clusters of MV permissive, intermediate, or resistant samples.

6. Also please add the patient labels to the bottom scatter plot of Figure 3E and provide a 95% CI of correlation coefficient.

7. On page 14, there is one erroneous paragraph break.

Reviewer #1 (Remarks to the Author): with expertise in oncolytic viruses, GBM

Comment 1: In this first in human trial of an oncolytic measles, the authors treat 23 patients with recurrent malignant gliomas via a 2 arm study: in arm A 3 dose-escalation of the agent was done via peritumoral injection, while in arm B an initial injection of the measles was done followed 5 days later by resection and peritumoral injection. The authors show safety of the agent. In addition, they provide important evidence of a 22 interferon gene signature profile inversely correlated with MV replication and infection in the arm B subjects where post-injection tumors were available. Additional, nanostring analyses clustered permissive and non-permissive tumors based on pathways associated with immune responses. In addition, there was also a correlation between increased tumor permissiveness to the MV and greater TILs in tumor. These findings are highly significant in that they correlate MV replication with increased TILs in a tumor such as malignant glioma, known to be lymphocyte-depleted. Therefore, the authors convincingly demonstrate that MV via its replication can alter the tumor microenvironment and enrich it in TILs. I have some minor comments primarily as a means of clarification and not as criticisms for this otherwise highly significant paper.

Response: We thank the reviewer for their positive comments.

Comment 2: Patients were on corticosteroids. Was there any effect of dose or time on steroids on tumor permissiveness or induction of TILs?

Response: We collected information on corticosteroid use at study entry. There was no significant difference in viral replication, as reflected by viral genome copy in post treatment biopsy, by corticosteroid use (yes/no) at study entry (supplemental figure 5A).

There was a significantly greater increase in percentage of CD4, CD8, and CD20 positive cells from pre to post treatment biopsies among patients who did not use corticosteroids at baseline as compared to those who did (supplemental figure 5B).

This information has now been included in the manuscript (sections on Molecular profiling of tumors in measles virus treated patients and Assessment of immune cell subpopulations in tumor specimens).

Comment 3: Table 2- what is the genome copies/ ug RNA of the input dose? I am trying to determine if there is an increase, decrease or no change in genome copies 5 days after injection.

Response: In order to respond to the reviewer's question, we used the baseline pre-virus treatment scan in Group B patients with a single dominant injected lesion to calculate tumor volume in order to estimate weight and μg of RNA in the injected tumor. We then divided viral copy numbers in the input dose with the estimated μg of RNA in the injected tumor to calculate input genome copies/ μg of RNA and compare with output (supplemental table 5). Remarkably, the two patients with the lowest DLDA score whose tumors were expected to be the most susceptible to viral replication, (pts 1 and 8), had the greatest increase between the day 1 input versus day 5 viral copy number/ μg of RNA output, again supporting replication and progeny formation. For intermediate DLDA score patients, results varied with stable or even decreased copy numbers in some patients indicating variable replication.

Comment 4: Figure 2- Since there was no CEA detected peripherally and MV antibodies did not rise, it seems that there is no spillage of the agent systemically after tumor injection. This provides further evidence of safety.

Response: We are in agreement with the reviewer's observation.

Comment 5: Figure 3- was there any correlation between the DLDA scores, MV receptor expression or the Nanostring analyses and survival?

Response: The Nanostring analysis was used to calculate the DLDA score (please see Methods). There was a non-significant association between DLDA score and survival ($p=0.43$, supplemental table 6).

There was no significant association between CD46, Nectin-4, SLAM and overall survival ($p=0.63$ for SLAM, $p=0.96$ for Nectin 4, $p=0.46$ for CD46, supplemental table 7).

Comment 6: Figure 4- can the authors divide up subject PFS and OS based on IDH mutation presence, 1p/19q deletion?

Response: We thank the reviewer for this comment: no study patients had 1p/19q deletion; two patients in Group A were positive for IDH mutation. Please see supplemental table 2.

Overall survival, but not progression-free survival, differed by IDH status. Median survival among IDH wildtype patients was 11 months (95% CI: 4.4 to 16 months); while for IDH mutant patients was 30 months (95% CI: 22 to NA months) (log rank p -value = 0.031). Median progression-free survival among IDH wildtype patients was 3.4 months (95% CI: 3.0 to 5.0 months); among IDH mutant patients was 4.8 months (95% CI: 3.0 to NA months) (log rank p -value = 0.650). Please see supplemental figures 7A, 7B, 7C and 7D. The above information has been added in the Results section of the manuscript.

Reviewer #2 (Remarks to the Author): with expertise in oncolytic viruses, GBM

Comment 1: Galanis et al. report the results of the FIH trial of MV-CEA in 22 patients with recurrent glioblastoma. The MV-CEA virus was constructed by introducing the extracellular N-terminal domain of human CEA in the measles virus backbone derived from the Edmonston vaccine lineage. The authors evaluate two different dosing strategies of the MV-CEA to assess the safety and preliminary efficacy. They also used the DLDA scoring system to predict MV permissiveness in GBM. The authors stated that patients with elevated baseline ISG expression, as reflected in a high DLDA score, had significantly lower levels of virus replication: a score above 150 indicates MV resistance, with scores below -250 associated with MV-permissive. An ISG-based DLDA algorithm could provide the basis for treatment personalization. This concept might help select patients who could benefit from MV-CEA treatment. It would be helpful to the readers to show serial magnetic resonance images (MRI) to assess the therapeutic effect of MV-CEA during and after intratumoral administration. For example, although the patients in Group B were candidates for gross total or subtotal resection, pre- and post-resection MRI would illustrate pre-dose tumor size, the extent of resection, and changes in contrast lesions or FLAIR high-signal regions associated with anti-tumor immune response.

Response: We thank the reviewer for this suggestion. We added supplemental figure 8 which includes a characteristic example of the evolution of imaging changes post treatment in a Group B patient.

Comment 2: Authors have previously shown that viral propagation correlates with CEA levels, which means they could monitor the activity of the virus by testing CEA levels in the peripheral blood. However, only one patient had an elevated MV-CEA in the peripheral blood after administration in this study. Please provide pre- and post-treatment CEA levels for all cases, as this will explain that CEA levels may have increased

partially.

Response: There was a slight increase in CEA levels from pre- to post-treatment. Absolute mean change in CEA levels from pre- to post-treatment overall was 0.127 ng/ml (95% CI: -0.093, 0.348, paired t-test p=0.244). The absolute mean change in CEA levels for Arm A was 0.089 ng/ml (95% CI: -0.256, 0.434, paired t-test p=0.569); for Arm B, 0.154 ng/ml (95% CI: -0.176, 0.484, paired t-test p=0.329). This additional information is now included in Results (in the CEA section). Pre and post treatment CEA levels for all patients are shown in supplemental figure 9.

Comment 3: The trial title states that it is for recurrent glioblastoma. However, the selection criteria are recurrent Grade 3 or 4 gliomas, including astrocytoma, oligodendroglioma, or mixed glioma with histologic confirmation at initial diagnosis or recurrence. IDH mutations are associated with better prognosis than IDH wild-type tumors of the same WHO grade (e.g., see <https://pubmed.ncbi.nlm.nih.gov/32557428/>). Initial diagnosis and IDH status should be included in supplemental tables.

Response: The phase I trial component allowed the enrollment of other high grade glioma patients; only 1 study patient had a grade 3 tumor, however. The manuscript title was adjusted accordingly. A total of 2 IDH positive patients were enrolled in the trial, both in arm A. Grade and IDH status are included as supplemental table 2. In addition, PFS and OS information for IDH wild type versus mutant patients are presented as supplemental figures 7A and 7B while in supplemental figures 7C and 7D we are presenting study PFS and OS with and without the inclusion of the 2 IDH positive patients. Because of the very small number of the IDH positive patients, the results are superimposable.

Comment 4: The authors also asserted that there was a moderate negative correlation between lower DLDA scores and a greater increase in CD4+ and CD8+ T cells following treatment. However, it is unclear whether CD4+ T cells increased after treatment, as the results were not shown. Please illustrate pre- versus post-treatment CD4+ T cell positivity and correlation with baseline DLDA scores in Group B patients. It would be more convincing if the authors could present representative pre- and post-treatment CD4+ and CD8+ T cell positivity of patients with scores above 150, below -250, and intermediate values (< 150 and > -250).

Response: Pre versus post CD4 levels and correlation with DLDA score are provided as supplemental figure 6A and 6B. Most patients had intermediate DLDA scores (between -250 and 150); there was only one patient with a DLDA score above 150 (see figure 3A and supplemental figure 10). Therefore, we chose to represent DLDA score on a continuous scale in these results.

Comment 5: The protocol states that adverse events should be graded at each evaluation, and pre-treatment symptoms and conditions should be evaluated at baseline. It is appropriate that Figure 1 illustrates treatment-related adverse events. However, since this is an FIH study of MV-CEA and patients will undergo en bloc tumor resection at the time of administration, unrelated adverse events occurring after MV-CEA administration should also be shown in a supplementary table.

Response: Please see detailed information regarding unrelated adverse events in supplemental table 8.

Comment 6: Discussion – The OS of patients with resectable recurrent GBM is relatively good. The details of the A071101 trial indicate that the OS is around 8 months for patients using HSPPC-96, but among patients treated per protocol, the OS for bevacizumab alone is 12.3 months. The authors states observed median OS of 11.5 months compares favorably with other contemporary studies in the recurrent GBM population. However, since the method of administration to demonstrate efficacy will be studied in the

future, it is better not to emphasize efficacy here.

Response: We agree with the reviewer that caution should be exercised in interpreting efficacy results in phase I/II trials. We report on PFS and OS as these represent the trial's secondary endpoints. The discussion has been revised to deemphasize efficacy and highlight the importance of prospective validation of these clinical findings.

Comment 7: Statistical analyses – progression-free survival was defined as the length of time from date of registration to a) date of progression or death due to any cause or b) last follow-up. However, on page 6, PFS was defined as the time from start of study therapy to documentation of disease progression. Which is correct? How long did it take from the registration date to the first dose of MV-CEA? The starting point of the survival curve would usually be the surgery date or the start of study therapy.

Response: Overall survival and progression-free survival were defined in the protocol as length of time from date of registration. We appreciate the reviewer's comment that typically survival curves start from start of study therapy. In this case, the curves are essentially superimposable. Patients started study therapy immediately after registration: median time from registration to treatment start was 2 days (range: 0 to 9 days) for arm A and 1 day (range: 0 to 8 days) for Arm B. PFS and OS outcomes were calculated from the registration date as defined in the protocol.

We have now clarified in Methods that PFS was defined as time from registration to documentation of disease progression.

Comment 8: Figure 2 – Comparison within each group shows no difference in antibody levels before and after treatment, but there is a significant difference between Group A and Group B.

Please provide evidence that there is no problem adapting 10⁷ TCID₅₀ determined to be MTD from Group A to Group B.

Response: Despite the difference in starting antibody levels, these levels per study eligibility were within the range that has been associated with protective systemic immunity against measles virus.

Regarding the reviewer's second sub comment, in consultation with the FDA and in order to address the possibility of unexpected toxicity with repeat administration, when we reached the highest dose of 10⁷ TCID₅₀ in Group A, we decreased the starting dose in Group B by 1 log per dose (10⁶ TCID₅₀) to be administered on days 1 and 5.

Safety results were very reassuring: In Arm A, two Grade 2 AEs were observed in the highest dose level of 10⁷ TCID₅₀ administered in the resection cavity. In Arm B, three Grade 2 AEs were observed at the lower dose level (Dose Level 10⁶ TCID₅₀ administered intratumorally day 1 and in resection cavity on day 5, i.e., a total dose of 2 x 10⁶ TCID₅₀). These Grade 2 events did not constitute dose-limiting toxicities and as such we proceeded with dose escalation to the highest dose level of 2x10⁷ TCID₅₀. No Grade 2+ AEs related to treatment were observed among the 10 patients treated at this highest dose level in Arm B and this was determined to be the MTD (supplemental figure 4).

Comment 9: Figure 3- Figure 3E shows a moderate negative correlation between lower DLDA scores and a greater increase in CD8+ T cells after treatment. However, adding patient numbers to the black dots in the bottom graph would be clearer to clarify the relationship with the patients in the top graph. The time point indicated by the pre-treatment baseline in Figure 3F needs to be clarified. The values where the dotted line

intersects the vertical axis differ significantly from the pre-treatment values in Figure 3E. For example, the percentage of CD8-positive cells before treatment for pt 07 is about 5% in Figure 3E, while the vertical axis in Figure 3F crossed with the dotted line is 20%. Please correct Figure 3F.

Response: Please see revised Figures 3E and 3F.

We agree with the reviewer that Figure 3F could be clarified. The original Figure 3F included, in addition, pre-study trends in CD8 infiltration in some patients who also had more than one surgery prior to enrollment, which we appreciate can be confusing. Since the goal of this analysis was to evaluate if treatment induced lymphocytic infiltration would persist over time at the absence of additional viral treatment, the figure was adjusted accordingly to start on study day 5, at which time the patients received the last viral dose. The results suggest reversal of the proinflammatory tumor microenvironment over time in the majority of patients.

Comment 10: Figure 4-Since DLDA scores and survival time for each patient are not shown in tables, it is difficult to know to what extent the DLDA scores affected survival in Figure 4. Please include in tables.

Response: Please see supplemental table 4 which examines the relationship between DLDA score and OS, as well as our response to Reviewer #1 Comment 5.

We did not observe an association between DLDA score and survival (supplemental table 6); this, however, could reflect the limited number of viral doses administered (maximum of 2 doses per patient per study design) with reversal of the proinflammatory remodeling of the tumor microenvironment in the majority of patients following study treatment completion, as figure 3F suggests.

Reviewer #3 (Remarks to the Author): with expertise in oncolytic viruses, GBM

Comment 1: This is an important clinical study from a team that has championed oncolytic virotherapy for glioblastoma. The authors have completed yet another human study which in and of itself is an incredible accomplishment. I find the work very important, solid, and in need of publication and dissemination. My remarks below are for consideration. Table 1. Could we gain some more information about IDH status or MGMT status of these patients?

Response: Two patients, both in group A, were IDH mutant. Please see response to Reviewer #1, comment 6, regarding overall survival and progression free survival by IDH status. The manuscript has been modified to include this information and supplemental table 2 and supplemental figure 7 were added.

Out of 22 patients, 20 had MGMT methylation status results available (please see supplemental table 2). One patient was MGMT methylated and had a longer survival outcome (21 months) as compared to the 19 MGMT methylation negative patients (median survival of 11 months [95% CI: 4.3, 18]), but the MGMT positive patient number was too small for a formal evaluation.

Comment 2: Table 2: Measles virus is detected by qPCR....do the authors have any data to support that these are viable and fully assembled viral particles? This would be helpful in the overall discussion.

Response: Vero cell overlays or progeny viral particle characterization was not performed. Nevertheless, the increase in viral copy number/ μ g RNA detected post administration, more prominently in permissive patients with low DLDA scores, supports viral replication. Please also see response to comment 3 of reviewer 1.

Comment 3: Figure 2: I find this somewhat interesting since we usually see antiviral immunity limit OV. It would appear that anti-MV antibody levels go down in both arms, though probably not statistically significant...how do the authors explain this phenomenon?

Response: We thank the reviewer for this comment. The decrease in pre versus post anti-MV antibody levels was very small and not statistically significant. In arm A there was a mean change of -6.7 units (95% CI: -17.2 to 3.9; paired t-test p-value=0.178). In arm B, the difference -2.7 units (95% CI: -6.1 to 0.76; paired t-test p-value = 0.112). This information has now been added to the manuscript.

Comment 4: Digging a little deeper...was the survival affected at all by IDH, MGMT expression?

Response: Please see response to comment 1 above, and supplemental figure 7.

Comment 5: Does the level of viral replication in patients correlate in any way with CD46 or nectin or SLAM?

Response: There was no significant correlation between viral replication and SLAM (Spearman's rho= -0.19, p= 0.56) or CD46 (Spearman's rho= -0.11, p=0.73) expression. There was a negative correlation between viral replication and Nectin-4 expression (Spearman's rho= -0.65, p=0.023). Please see supplemental figure 11.

Comment 6: Figure 3C...the differential gene expression is interesting...have the authors validated any of these targets at the protein level?

Response: The DLDA signature primarily consists of interferon stimulated genes (ISG). ISG are not expressed on the cell surface and as such it is difficult to detect at the protein level. We have, however, performed some parallel work to further study specific ISGs that mediate restriction of measles virus (MV). For example, we have reported that Radical S-adenosyl methionine domain containing 2 (RSAD2-one of the ISGs included in the 22 gene DLDA calculation) restricts measles virus infection at the stage of virus release. We have also tested the impact of RSAD2 expression for oncolytic virotherapy using the measles virus permissive line SR-B2: measles virus release was impaired in SR-B2 cells transduced to express RSAD2 in vitro. Additionally, oncolytic MV therapeutic efficacy was impaired in SR-B2 cells transduced to express RSAD2 in vivo (Kurokawa C et al. A key anti-viral protein, RSAD2/VIPERIN, restricts the release of measles virus from infected cells. *Virus Res* 2019; 263:145-50.)

Comment 7: Besides CD8 analysis...is there are information of CD4 levels or Tregs?

Response: In addition to CD8, we also analyzed pre- and post-tumor tissues for expression of CD4, CD3, CD20 and CD68. Supplemental figure 6A and 6B was added, depicting pre- versus post-treatment CD4+ T cell infiltration and correlation with baseline DLDA score in Group B patients. Treg cells were not specifically evaluated.

Comment 8: Statement in the discussion suggests that MV-CEA is associated with biological activity despite the fact that 41% of patients had received chemo...can the authors expand on their thinking here? I am not sure what the message is and this is probably more me than them.

Response: Given the correlative analysis results demonstrating viral replication and proinflammatory remodeling of the tumor microenvironment, we use the term biologic rather than clinical activity to try and

soften our efficacy conclusions since efficacy was a secondary endpoint in this trial.

Reviewer #4 (Remarks to the Author): with expertise in biostatistics, bioinformatics

Comment 1: The authors published a JNCI paper in 2018 about the safety and efficacy of a novel oncolytic Measles virus (MV) derivative in patients with glioblastoma based on a preliminary data of their ongoing clinical trial. In addition, they built a 22-gene signature for predicting MV permissiveness and validated it on subjects participated up to that time.

Based on the final analysis dataset of the clinical trial with three additional subjects, the current manuscript presents an updated result of safety/efficacy of the MV derivative as well as a validation result of the 22-gene signature.

The result showed that the MV therapy is safe with no severe adverse events and has a comparable efficacy to contemporary treatments for glioblastoma. There was a significant increase in CD8+ and CD68+ T cell infiltration in 11 patients' tumors. The 22-gene signature score has a little weakened but still significant negative correlation ($r=-0.6$, $p=0.04$) to the degree of MV replication. Overall, the work is well executed and statistically sound. I have minor comments as follows:

Regarding Study Design section on page 4 and 5, it appears that patient group A and B are from the 3+3 design cohort or the dose expansion cohort, respectively. However, the group were not defined in the study design section. Please clarify it in the manuscript and the trial schema (the supplementary Figure 1).

Response: Both treatment arms used a cohort of 3 design to determine the MTD for both treatment schedules. Group A included only dose escalation. Group B included both dose escalation and a dose expansion cohort.

We have clarified the study design in the Methods section as follows:

"This single-arm, phase I/II trial enrolled patients to two treatment regimens in a standard 3+3 cohort design, with additional patients to be enrolled in a maximum tolerated dose (MTD) expansion cohort. Cohorts of 3 patients were accrued in Group A until the MTD of MV-CEA was determined following single dose administration. Subsequently, cohorts of 3 patients were enrolled in Group B for further evaluation of the MTD following administration of two viral doses. Additional patients were then enrolled to the Group B MTD expansion cohort. The study schema is summarized in Supplemental **Figure 1.**"

Comment 2: In DLDA section on page 8, two cutoffs of -250 and 150 for DLDA score were used to classify MV resistance. However, there is no description about how those cutoffs were derived.

Response: We have described the development of this DLDA model in a previous manuscript (Kurokawa et al. JNCI, 2018). The DLDA model was trained on gene expression data from GBM PDX lines known to be MV permissive or resistant. Coefficients and a constant from the training set DLDA model were used to calculate DLDA prediction scores for subsequent samples, including GBM and ovarian cancer PDX xenografts samples and the GBM patient samples, with a score above zero predicting MV resistance and a score below zero indicating MV permissiveness.

The cutoff scores were first described in the Kurokawa et al, JNCI, 2018 manuscript using data from 9 GBM patients with available viral replication values. To summarize, we observed that an increased DLDA score corresponded to decreased viral replication in tumor tissue. Three classifications emerged in relation to viral

replication. The patient with the highest DLDA score (>150) had no detectable virus, whereas the 2 patients with the lowest DLDA scores (< -250) had the highest level of virus recovered from the tumor. Furthermore, six patients clustered with moderate DLDA scores (-250 to +150) and intermediate levels of virus replication. The intermediate patients were permissive to virus replication; however, virus replication was approximately 2-log lower than the group with highest permissiveness. Therefore, GBM patient sample test-set DLDA scores above 150, between 150 and -250, and below -250 were classified as resistant, intermediate, and permissive, respectively.

Comment 3: Regarding Figure 1 on page 10, trial participants were treated at different doses of the MV therapy. Were there any difference in AEs between dose levels? I suggest stratifying them by dose levels and present or compare adverse events accordingly.

Response: In Arm A, two Grade 2 AEs were observed in the highest dose level (Dose Level 10^7 TCID₅₀ in resection cavity day 1). In Arm B, three Grade 2 AEs were observed at the lower dose level (Dose Level 10^6 TCID₅₀ intratumoral day 1 and in resection cavity day 5). However, these Grade 2 events did not constitute dose-limiting toxicities. Notably, no Grade 2+ AEs related to treatment were observed among the 10 patients treated at the highest dose level in Arm B.

To address the reviewer's comments, we have added supplemental figure 4 and supplemental tables 4 and 9, and have made the following changes in the text (in the Results section "Safety" paragraph):

In total, 14 patients (63.6%; Group A: 7 patients [77.7%]; Group B: 7 patients [53.8%]) reported a treatment-related adverse event (TRAE). Overall, 4 patients reported a Grade 2 TRAE: fatigue was reported by 2 patients [1 patient in Group A (10^7 TCID₅₀) and 1 patient in Group B (2×10^6 TCID₅₀)], 1 patient had anemia [Group A (10^7 TCID₅₀)] and 1 patient in [Group B (2×10^6 TCID₅₀)] reported both lymphopenia and speech impairment (supplemental figure 4 and supplemental table 4).

Comment 4: In Molecular profiling of tumors in MV-treated patients section on page 11, DLDA score has a significant negative correlation, but it largely depends on a few extreme observations such as Pt1 and Pt10. Please provide a 95% interval of the correlation coefficient. It is also suggested to make 3 groups of subjects by using the two DLDA cutoffs (< -250, intermediate, > 150) and then compare the degree of virus replication. This could be another way to validate the DLDA score and justify the relevance of DLDA cutoffs.

Response: We chose to represent DLDA as a continuous variable as most observations were within the intermediate DLDA category.

The one patient with the highest DLDA score (equal to or above 150) had no detectable virus. The mean viral genome copy number post treatment for the 8 patients with intermediate DLDA scores (above -250 and lower than +150) was 5.02×10^4 (range: 1.3×10^3 - 1.7×10^5 MV genome copies/ μ g of RNA). Mean viral genome copy number for the 3 patients with low DLDA scores (less than or equal to -250) was 2.23×10^7 (range: 1.6×10^3 to 6.0×10^7 MV genome copies/ μ g of RNA).

Supplemental figure 10A depicts the DLDA/viral replication correlation with the DLDA cutoffs marked, and supplemental figure 10B shows the viral genome copies per DLDA group. Mean and median genome copies were approximately 3 logs higher in the low versus intermediate score group, further supporting the biologic relevance of the proposed cutoffs.

The 95% correlation coefficient has been added to figure 3A.

Comment 5: On page 12, it is hard to follow that “We observed ... permissive tumors formed one cluster (right side of heat map in Figure 3C) Resistant formed a cluster on the opposite branch of the hierarchical tree (left side of heat map in Figure 3C)”. Please label samples or add marks in Figure 3 that will help distinguish clusters of MV permissive, intermediate, or resistant samples.

Response: Figure 3C and corresponding legend was revised as recommended by the reviewer: “Permissive tumors are underlined with black bar, intermediate – with green bar and resistant with magenta bar at the bottom of the heatmap.”

Comment 6: Also please add the patient labels to the bottom scatter plot of Figure 3E and provide a 95% CI of correlation coefficient.

Response: Figure 3E has been updated with patient labels. The text has been updated with the Spearman correlation coefficient for CD8: -0.46 (95%CI: -0.83 to 0.19, $p = 0.154$).

Comment 7: On page 14, there is one erroneous paragraph break.

Response: This has been corrected.

REVIEWERS' COMMENTS

Reviewer #1 (Remarks to the Author):

The authors have answered all my queries.

Reviewer #2 (Remarks to the Author):

The authors provide a revised manuscript. The changes made enhance the quality of the paper, making it a solid contribution to the oncology literature. I have no further comments and recommend the manuscript for publication.

Reviewer #3 (Remarks to the Author):

Thank you for addressing all my concerns.

Reviewer #4 (Remarks to the Author):

All my questions and comments are addressed well along with adequate supplementary figures and tables. I thank the authors and have no further comment.